# Spatially confined lignin nanospheres for biocatalytic ester synthesis in aqueous media

Mika Henrikki Sipponen [1], Muhammad Farooq[1], Jari Koivisto[2], Alessandro Pellis [3], Jani Seitsonen[4] & Monika Österberg [1]

Dehydration reactions proceed readily in water-filled biological cells. Development of bio-catalysts that mimic such compartmentalized reactions has been cumbersome due to the lack of low-cost nanomaterials and associated technologies. Here we show that cationic lignin nanospheres function as activating anchors for hydrolases, and enable aqueous ester synthesis by forming spatially confined biocatalysts upon self-assembly and drying-driven aggregation in calcium alginate hydrogel. Spatially confined microbial cutinase and lipase retain 97% and 70% of their respective synthetic activities when the volume ratio of water to hexane increases from 1:1 to 9:1 in the reaction medium. The activity retention of industrially most frequently used acrylic resin-immobilized *Candida antarctica* lipase B is only 51% under similar test conditions. Overall, our findings enable fabrication of robust renewable bio-catalysts for aqueous ester synthesis, and provide insight into the compartmentalization of diverse heterogeneous catalysts.

[1] Department of Bioproducts and Biosystems, School of Chemical Engineering, Aalto University, PO BOX 16300, FI-00076 Aalto Espoo, Finland. [2] Department of Chemistry and Materials Science, School of Chemical Engineering, Aalto University, PO BOX 16100, FI-00076 Aalto Espoo, Finland. [3] Department of Chemistry, Green Chemistry Centre of Excellence, University of York, Heslington, York YO10 5DD, UK. [4] Nanomicroscopy Center, Department of Applied Physics, School of Science, Aalto University, PO BOX 11000 , FI-00076 Aalto Espoo, Finland. Correspondence and requests for materials should be addressed to M.H.S. (email: mika.sipponen@aalto.fi) or to M.Ö. (email: monika.osterberg@aalto.fi)

Enzymatic catalysis continues to open new pathways to site and stereoselective transformation and synthesis of bioactive substances, fine chemicals, and polymers[1,2]. One of the advantages of biocatalysis is the possibility to use mild reaction conditions and aqueous reaction media. In contrast to the compartmentalized ester synthesis within water-filled biological cells, essentially all of the literature on immobilization of industrial hydrolases such as microbial lipases[3] and cutinases[4] involves physical or chemical surface-binding to synthetic materials[5–10]. Adsorption on solid carriers increases the activity and stability of lipases[11,12], but it also exposes enzymes to water. This has been a major hurdle for aqueous synthesis of short-chain esters.

In turn, water-immiscible substrates have prevailed in lipase-catalyzed ester synthesis in microaqueous[13,14], aqueous biphasic[15], and aqueous emulsion[16–18] media. However, the ability to synthesize short-chain esters and polymers in water could potentially revolutionize industrial biocatalysis. For instance, esterification of organic substances arising from thermochemical biomass conversion and microbial fermentations would facilitate phase separation and circumvent the energy-intensive distillation of dilute aqueous solutions. Recent reports on the spatial confinement of proteins[19–21] encourage development of compartmentalized biocatalytic materials for condensation reactions in the presence of water.

Here, we address this challenge by fabricating spatially confined biocatalysts upon self-assembly and drying-driven aggregation of enzyme-cationic lignin nanosphere (c-CLP) complexes in calcium alginate hydrogel. We use colloidal lignin particles (CLPs) as low-cost enzyme carriers because these spherical nanoparticles offer large specific surface area for adsorbing proteins[22], their scalable production does not require chemical modification[23], and physical adsorption suffices for surface functionalization[22,24]. Contrary to earlier reports[9,25–27], our immobilization method does not require covalent cross-linking to the support or synthetic polymers to stabilize lipases. Our results show that the spatially confined Lipase M from *Mucor javanicus* and *Humicola insolens* cutinase (HiC) exhibit unique efficiency and stability in the synthesis of butyl butyrate under a marked water excess.

## Results

**Fabrication of spatially confined biocatalysts**. The procedure for enzyme immobilization (Fig. 1a) began with a two-step adsorption process starting from CLPs. Detailed characterization of original and modified softwood kraft lignin (SKL) used in this work is available in the Supplementary Note 1 (see Supplementary Tables 1, 2 and Supplementary Figs. 1–3). A typical appearance of an aqueous CLP dispersion and single CLP particles (Z-average particle diameter = $177 \pm 2$ nm, $PdI = 0.10$) is shown in Fig. 1b–d while the $\zeta$-potentials as a function of pH for the various compounds are shown in Fig. 1e–f. In the first step, water-soluble cationic lignin (Catlig, $\zeta$-potential of $+19$ mV) adsorbed on CLPs ($\zeta$-potential of –38 mV). In the second step, the cationic as well as the aromatic moieties of c-CLPs adsorbed 91% (Supplementary Fig. 4) of the available Lipase M ($M_w = 33$ kDa, Supplementary Fig. 5). The process was completed by entrapping the enzyme-c-CLP complexes in calcium alginate and drying at room temperature, forming durable beads that did not re-swell in aqueous buffer solutions, except when using 0.2 M sodium phosphate buffer at pH 7 (Supplementary Fig. 6). Electron microscopy images revealed clusters of cationic lignin nanospheres at cross sections of dried beads (see Supplementary Fig. 7 in Supplementary Note 3). The overall mass balance showed that the yield of the alginate beads was $90 \pm 8\%$ on dry weight basis (Supplementary Table 4), while the protein mass balance

indicated an enzyme immobilization efficiency of 96% (Supplementary Fig. 4). Alterations of this basic procedure consisted of entrapping Lipase M in calcium alginate alone or with various surface-active materials.

**Biocatalytic synthesis of butyl butyrate in aqueous media**. Butanol and butyric acid are well-suited substrates to assess the aqueous esterification reaction because unlike these water-soluble metabolites of bacterial fermentation, the produced butyl butyrate is insoluble in water. The adsorption-entrapment immobilization method described above was found superior when compared to our initial attempts with layer-by-layer immobilized Lipase M (see Supplementary Figs. 8, 9 in Supplementary Note 4). The molar yield of butyl butyrate reached 77% in 96 h in the reaction catalyzed by spatially confined Lipase M (adsorbed on c-CLPs prior to the entrapment in calcium alginate). Products accumulated linearly up to 24 h reaction time, while increasing the catalyst dosing by a factor of three increased the 48 h molar yield by only 19% (Supplementary Fig. 9c). Importantly, the reaction yield after the first 24 h reaction (37%) was three times as high as that obtained when alginate beads contained Lipase M alone (Fig. 2a). Further, in the presence of c-CLPs there was a mere 3% reduction in the synthetic activity after the first reaction step compared to a drastic activity loss of 41% without c-CLPs.

Our hypothesis was that the previously found beneficial amphiphilic properties of c-CLPs in emulsion stabilization[24] were important in the present work as well. To assess activation by surfactants[28–30], we entrapped Lipase M in calcium alginate with Catlig, nonionic Tween 80, anionic SDS, and cationic didodecyldimethylammonium bromide (DDMA) surfactants (0.6% of the dry weight of alginate beads and 10% relative to the dry weight of lipase M). Tween 80 performed best among the surfactants, but after two repeated 24 h reactions the reaction yield dropped to a half of that achieved with the beads containing Catlig (Fig. 2b). SDS inhibited Lipase M, as previously found in olive oil hydrolysis[24]. The superior reaction yields obtained with the biocatalysts containing Catlig suggests that its aromatic three-dimensional structure with multiple cationic and anionic sites is beneficial (Fig. 2c). The synthetic reaction was carried out at pH 3, and under these conditions c-CLPs are cationic ($\zeta$-potential of $+15$ mV, Fig. 1c) while Lipase M is slightly anionic ($\zeta$-potential of $-2.2$ mV). Therefore, c-CLPs are feasible anchors for the entrapped anionic hydrolases, retaining them in the solid matrix. The use of Bu-CLPs instead of regular CLPs, or replacement of one third of the amount of CLPs with hydrophobic spherical Carnauba wax particles[31], did not increase the esterification activity (Supplementary Fig. 10a). Probably the particle size is more important. CLPs were smaller than Bu-CLPs or wax particles (Supplementary Fig. 10d) and thus provided a larger net surface area for lipase immobilization.

To generalize the benefits of the developed immobilization method, we entrapped microbial cutinases (HiC) in alginate beads with c-CLPs or Tween 80. These biocatalysts gave equally high 73–77% molar yields of butyl butyrate in 24 h biphasic reactions (Supplementary Fig. 11). It is also important to minimize the amount of organic solvent in the aqueous ester synthesis during a reactive extraction process. For this, we determined the retention of synthetic activity when the water content of the reaction medium was significantly increased. The enzymes were dosed based on their hydrolytic activities (see Supplementary Note 2, Supplementary Table 3). Spatially confined HiC and Lipase M retained 70 and 97% of their synthetic activities, respectively, upon increasing the volume ratio of water to hexane from 1:1 to 9:1 (Fig. 2d). These values are considerably higher than the 51% activity retention of acrylic

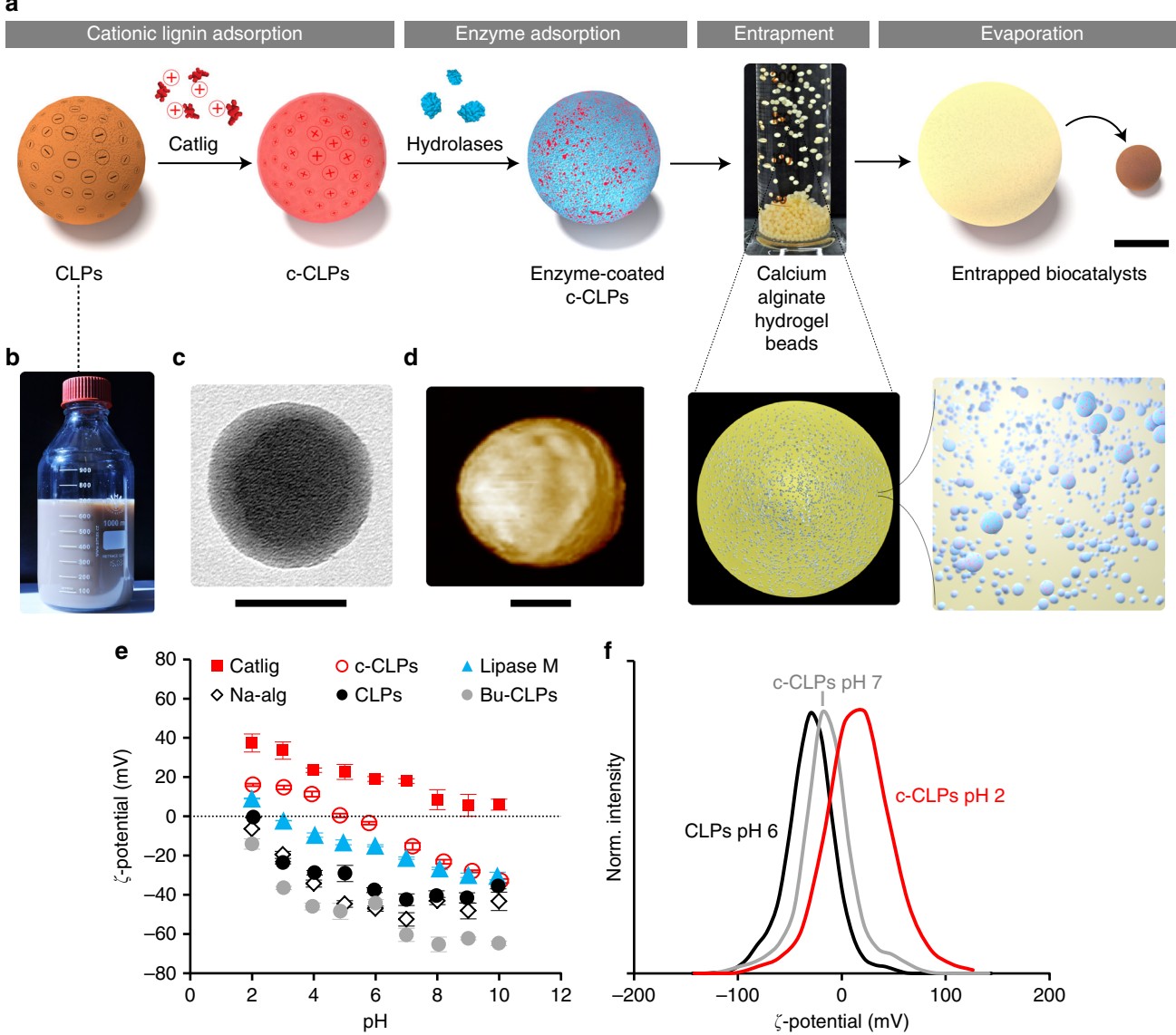

**Fig. 1** Spatial confinement of enzymes utilizing colloidal lignin particles. **a** Scheme of the enzyme immobilization process comprising (i) Adsorption of cationic lignin on CLPs to yield c-CLPs. (ii) Adsorption of hydrolases on c-CLPs. (iii) Entrapment of enzyme-c-CLP complexes in calcium alginate hydrogel, and (iv) Consolidation of the alginate beads by evaporation of water under ambient conditions. **b** Photographic, **c** TEM, and **d** AFM images of CLPs. Scale bars: **a** = 1 mm, **c**, **d** = 50 nm. **e** $\zeta$-potentials of the biomaterials as a function of pH. Error bars represent two standard deviations from the mean values (three measurements on a single replicate, except for Catlig six measurements on two replicates). **f** $\zeta$-potential distributions of CLPs and c-CLPs

resin-immobilized CALB, the state-of-the-art industrial lipase, under the corresponding conditions. These results represent an important step towards circumventing the thermodynamically disfavoring effect of water in aqueous-phase esterification[32,33].

In contrast to the present work achieving a 52% molar yield with spatially confined HiC in 24 h in the presence of the 90% volume fraction of water, prior studies in aqueous biphasic, microaqueous, or anhydrous reactions have used anhydrous organic solvents or solvent mixtures with much less water (Table 1). Yet, the reaction yields of the present work compare favorably to the earlier data. For instance, the 24 h reaction yield was 91% in hexane-water 1:1 v/v when using 80 mg (dry basis) of reusable spatially confined biocatalyst (HiC with c-CLPs, see Supplementary Table 4). Repetition of the reaction after 8 months storage of the immobilized enzymes at 4 °C gave a butyl butyrate yield of 64% (Supplementary Fig. 12a). Another benefit offered by our immobilization system is the stabilization of lipases under

reaction conditions. The time required to lose one half of the catalytic activity ($t_{1/2}$, Supplementary Fig. 10e) under actual reaction conditions is higher with spatially confined Lipase M (98 h) and HiC (161 h) compared to the literature values of 24–32 h with covalently immobilized lipases[27,34]. Furthermore, the ability to carry out the reaction in the presence of the 90% volume fraction of water further stabilized Lipase M ($t_{1/2} = 110$ h).

**Cationic lignin nanospheres as templates for hydrolases.** Figure 3 shows schematic models and associated microscopic images of different approaches for Lipase M immobilization; with CLPs, Catlig, or c-CLPs. In the first case, adsorption on spherical lignin particles is, as such, beneficial for the activation of hydrolases, but the enzyme-coated CLPs did not form compartmentalized assemblies in alginate beads (Fig. 3a and Supplementary Fig. 5c). In the second case, water-soluble cationic lignin

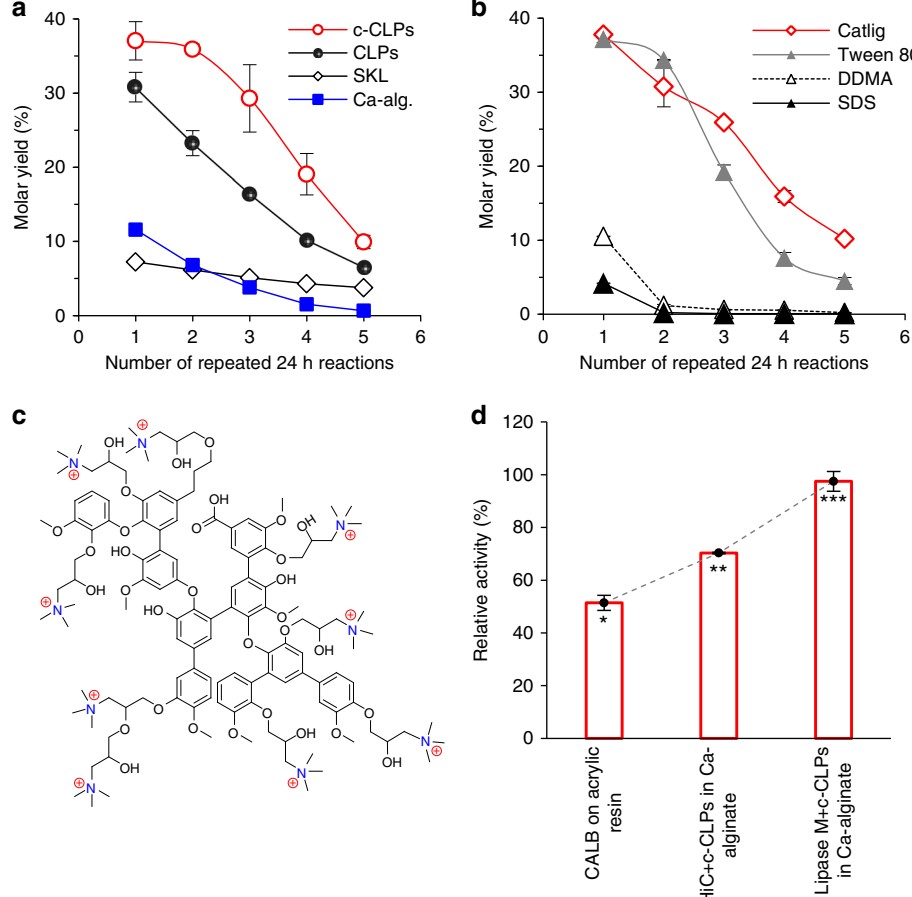

**Fig. 2** Enzymatic butyl butyrate synthesis. **a**, **b** Five repeated 24 h reactions catalyzed by Lipase M entrapped in calcium alginate alone or with CLPs, c-CLPs, SKL, water-soluble cationic lignin (Catlig), Tween 80, DDMA, or sodium dodecyl sulfate (SDS). **c** A structure model of cationic lignin[24]. **d** Relative synthetic activity (the ratio of synthetic activity in hexane-water 1:9 v/v to the activity detected in hexane-water 1:1 v/v) of immobilized hydrolases. The reaction conditions in **a**, **b** and **d** were similar (40 °C, pH 3, 200 rpm agitation). Error bars represent two standard deviations from the mean values (two replicates on a single batch of biocatalyst, except for c-CLP in a six replicates (three biocatalyst batches) for the first reaction, and four replicates (two biocatalyst batches) for the reactions 2–5). Asterisks in **d** indicate statistically significant differences ($p < 0.05$, one-way ANOVA with Tukey's HSD test). Details of ANOVA associated with **a**, **b** are given in Supplementary Table 5

**Table 1 Comparison of reaction conditions and yields of enzymatic butyl butyrate synthesis**

| Reaction system[a] | Enzyme[b] | Reaction | | Ref. |
|---|---|---|---|---|
| | | Time (h) | Yield (%) | |
| O (heptane) | Immob. *Candida rugosa* lipase | 48 | 69 | 25 |
| O (hexane) | *C. cylindracea* lipase (free or immob.) | 91 | 87 | 14 |
| O (heptane) | Immob. *C. rugosa* lipase | 48 | 69 | 27 |
| B (90%) | Immob. HiC | 24 | 52 | This work |
| B (50%)[c] | Immob. HiC | 24 | 91 | This work |
| B (50%) | Immob. Lipase M | 96 | 77 | This work |
| B (5%) | *R. miehei* lipase | 6 | 100 | 13 |
| Aq. | Immob. *C. rugosa* lipase | 96 | 32[d] | 47 |

[a]O, organic anhydrous (solvent); B biphasic (% of water); Aq., aqueous
[b]Immob., immobilized
[c]Fourfold enzyme dosing compared to that in the reaction in 90% water content
[d]Detected after rigorous extraction that destroyed the biocatalyst

co-precipitated with lipases due to the electrostatic neutralization of surface charges (Fig. 3b). The entrapped Catlig-Lipase M precipitates were observed in the scanning electron microscopy (SEM) and confocal laser scanning microscopy (CLSM) images as free and nested clusters (Fig. 3g–j). The third and most beneficial immobilization approach builds on the positive traits of the two

aforementioned systems (Fig. 3c). A key phenomenon is the self-assembly of enzyme-carrying c-CLPs in the calcium alginate hydrogel. CLSM images of acridine orange-stained sections confirmed enrichment of protein in the nanosphere assemblies (Fig. 3k–m) also observed in the SEM images (Fig. 3l).

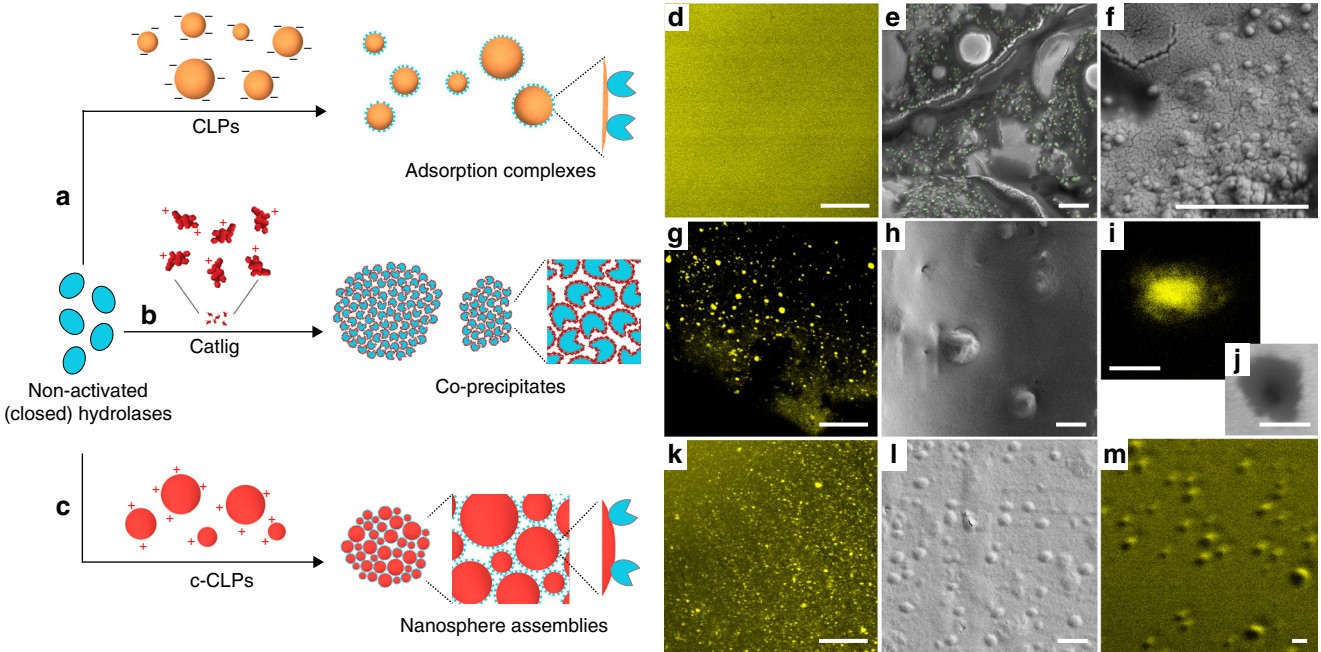

**Fig. 3** Models and microscopic images of immobilized hydrolases. **a** Adsorption of hydrolases on colloidal lignin particles (CLPs). **b** Formation of precipitates from enzymes and cationic lignin (Catlig). **c** Enzyme adsorption on cationic lignin nanospheres (c-CLPs), and formation of spherical clusters from enzyme-c-CLP complexes. CLSM images (**d**, **g**, **i**, **k**, **m**) associated with the materials in rows (**a**–**c**) show distribution of protein on acridine orange-stained cross-sections. SEM images show distribution of CLPs (**e** with green coloring, **f**), Catlig-Lipase M precipitates (**h**, **j**), and clustered c-CLPs and Lipase M (**l**). Scale bars: **d**, **g**, **k** = 40 μm, rest = 1 μm

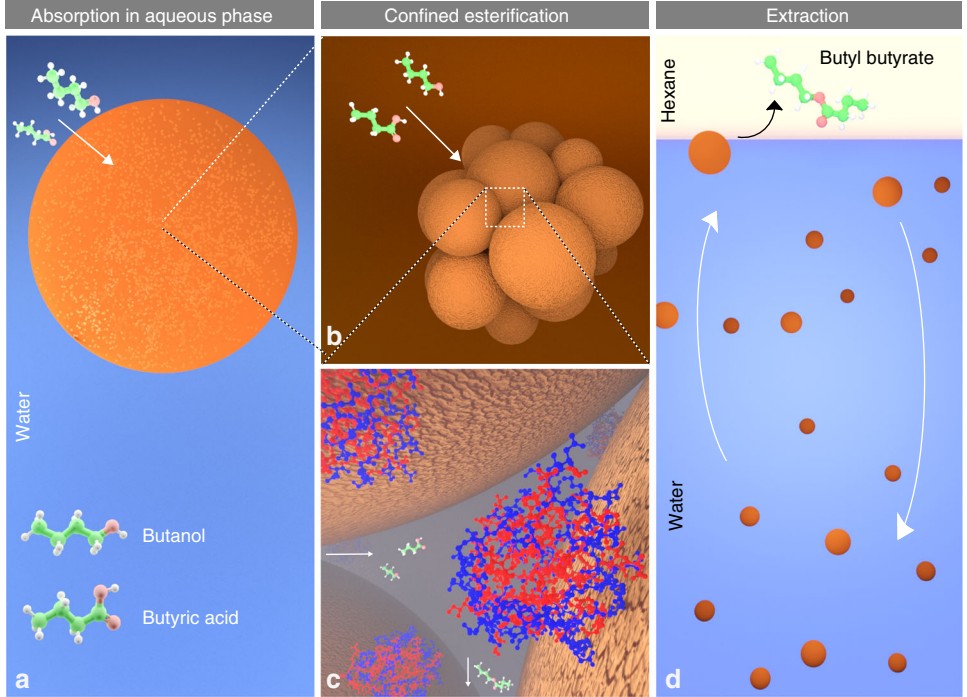

**Fig. 4** Schematic mechanism of the compartmentalized butyl butyrate synthesis. **a** The reagents diffuse from water into an alginate bead containing spatially confined enzymes. **b** Diffusion of the reagents into the biocatalytic clusters consisting of enzymes adsorbed on cationic lignin nanospheres. **c** Ester synthesis occurs in the inner space confined by the c-CLPs. Hydrolases act as gatekeepers that enhance the selective permeation of butanol and butyric acid. Hydrophobic residues are shown in red color for the HiC molecules[46]. **d** Diffusion of the produced butyl butyrate into the organic phase is driven by agitation that sequentially brings the biocatalyst beads to the solvent interface

The schematic mechanism in Fig. 4 visualizes the esterification reaction catalyzed by spatially confined biocatalysts. First, butanol and butyric acid diffuse into the alginate bead containing immobilized enzymes. Enzymes confined between adjacent nanospheres serve as gatekeepers for the organic reactants. The ability of the beads to resist swelling in water (Supplementary Fig. 6) means that laborious water removal by molecular sieves[35], distillation[13], or membrane reactors[36] becomes obsolete, while

the surface-bound water maintained structurally important hydration shell of lipases[37]. Low aqueous solubility of the synthesized butyl butyrate[13] facilitates its diffusion into the external organic phase during agitation, which drives the reaction toward esterification. Besides butyl butyrate, other short chain esters such as methyl butyrate[15] and butyl acetate[38] can be produced enzymatically using lipases, permitting the recovery of a plethora of bio-sourced molecules from water.

## Discussion

Here we have developed spatially confined biocatalysts that enable ester synthesis in the presence of water as the major component of the solvent system. This conceptually new immobilization process represents a significant step towards enzyme-catalyzed reactive extraction of organics from dilute aqueous streams. The scalable and environmentally benign spatially confined biocatalysts may also pave the way for next generation aqueous-phase polymerization processes.

## Methods

**Enzyme immobilization and synthesis of nanomaterials**. All lignin materials used in this work were prepared from BIOPIVA™ 100 pine kraft lignin (UPM, Finland). Cellulose nanofibril (CNF) hydrogel, CNF thin films, cationic lignin, CLPs, and carnauba wax particles (CWPs) were prepared and purified as described earlier[24,31,39]. Immobilization of hydrolases (Amano Lipase M from *Mucor java-nicus*, Sigma-Aldrich #534803 and purified[40] *Humicola insolens* cutinase (HiC) Novozymes, Beijing, China) started by adsorption of 9% (w w$^{-1}$) of cationic lignin on CLPs to render them cationic[24]. The second step involved adsorption of hydrolases onto the cationic lignin nanospheres. After the two 15 min adsorption steps at 22 °C with magnetic stirring at 150 rpm, the dispersion was mixed to homogeneity with aqueous sodium alginate (BDH Chemicals Ltd, UK). The colloidal mixture containing 2% (w w$^{-1}$) sodium alginate was precipitated dropwise to 1.25 volumes of 0.2 M aqueous calcium chloride at room temperature to obtain beads that were washed with deionized water and dried under ambient conditions. Alterations of this basic procedure included immobilization of Lipase M alone, with surfactants, dissolved or colloidal lignin, or CWPs with and without cationic lignin.

**Microscopy and particle analysis**. Atomic force microscopy (AFM) images of CNF thin films were captured as described in the literature[24]. For transmission electron microscope (TEM) imaging, 100 nm thick sections were cut from the alginate beads (containing Lipase M with various lignin materials) at a temperature of −100 °C using a Leica EM UC7 ultramicrotome. The thin sections were imaged using JEOL JEM-3200FSC instrument. Alginate beads embedded in epoxy resin were cut with a diamond blade to expose their cross sections for field emission-SEM (FE-SEM) and CLSM imaging. Gold/palladium sputtered cross-sections were imaged using a Zeiss Sigma FE-SEM operated at an accelerating voltage of 1.5 kV. Non-sputtered cross sections were imaged after acridine orange staining using a Leica DMRXE confocal laser scanning microscope with HCPL APO 20 × /0.70 CS and HPX PL APO 63 × /1.30 objectives. Particle size distributions and ζ potentials of various materials used in this work were analyzed in three replicates by dynamic light scattering using a Malvern Zsizer Nano-ZS90 and Mastersizer 2000 instruments.

**Characterization of enzymes and lignin**. Lipase M was solubilized at 10 g l$^{-1}$ in PBS buffer for SDS-PAGE analysis. The running gel contained 12% acrylamide. Hydrolytic activity was assayed in a mixture of 3 ml of tributyrin (#ACRO150882500, VWR), 2.5 ml of deionized water, and 1 ml of aqueous 200 mM Tris-HCl buffer (pH 7.0). After 30 min vigorous stirring at 40 °C, the reactions were stopped by adding 3 ml of 95% ethanol, and the reaction vials were transferred into ice. The released butyric acid was titrated using 0.1 M NaOH with phenolphthalein as an indicator.

Gel permeation chromatography of acetylated SKL followed published methodologies[24]. ATR-FTIR spectra of SKL, acetylated SKL, bytyrated SKL, and Catlig were recorded using a Mattson 3000 FTIR spectrometer (Unicam) with a GladiATR unit (PIKE Technologies). A total of 16 scans were recorded at a resolution of 2 cm$^{-1}$ and bands in the region 500–1800 cm$^{-1}$ were assigned[41]. Quantitative $^{13}$C and $^{31}$P NMR spectroscopy methods[42,43] were used to analyze structural moieties of lignin. HSQC spectrum of acetylated SKL (75 mg) was acquired in 0.5 ml of DMSO-d6 (#151874, Sigma-Aldrich). The Bruker pulse sequence *hsqcetgpsisp.2* was used to collect 96 scans at spectral widths of 12.2 and 242 ppm, for the $^{13}$C and $^{1}$H nuclei, respectively. $^{13}$C-$^{1}$H correlation signals were assigned according to literature[44,45].

**Enzymatic esterification reactions**. Procedures used in aqueous esterification reactions are given in the Supplementary Note 5. Biphasic esterifications were carried out in air-tightly capped glass vials containing the immobilized biocatalyst, n-hexane and aqueous solution of 0.15 M n-butanol and 0.05 M n-butyric acid. For the reactions in 1:1 solvent ratio, 2.5 ml of each liquid was used, and for the 1:9 volume ratio, 1 ml of hexane and 9 ml of the aqueous reactant solution were used, while maintaining the weight ratio of biocatalyst to reactants constant. The reaction vials were maintained at 40 °C and 200 rpm agitation in an orbital shaker. GC-FID analysis of reactants and products was made relative to isobutyl alcohol (i.s.) and utilized DB-WAXetr (30 m, 0.32 mm i.d., 1 μm film thickness) column from Agilent Technologies Inc., heated at 30 ml min$^{-1}$ from 80 to 200 °C. The injection volume was 0.5 μl and the split ratio was 1:20. One-way ANOVA and Tukey's HSD posthoc test ($n = 2$, except for c-CLP $n = 4$-6) were performed in Statistix 9 to deduce statistically significant differences ($p < 0.05$) in data presented in Fig. 2a, b, d (see Supplementary Note 6).

**Data availability**. The GC-FID, $^{31}$P NMR, and $^{13}$C NMR data that support the findings of this study are available in figshare repository at https://doi.org/10.6084/m9.figshare.5552584. Other data is available from the authors upon reasonable request.

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

## Acknowledgements
We thank Dr. Antti Nyyssölä (VTT), Dr. Ossi Pastinen, and Prof. Eero Kontturi (Aalto University) for critical discussions, M. Sc. Fei Xie-Sipponen for assistance with 3D model construction, and Dr. Alain Roussel and Dr. Christine Kellenberger (CNRS) for SDS-PAGE analysis. M.H.S and M.Ö. acknowledge Academy of Finland for funding (grants 296547 and 278279, respectively). A.P. thanks the FWF Erwin Schrödinger fellowship (grant agreement pJ 4014-N34) for financial support. Dr. Suvi Pietarinen (UPM) is thanked for providing SKL. We acknowledge the provision of facilities and technical support by Aalto University at OtaNano - Nanomicroscopy Center (Aalto-NMC).

## Author contributions
M.H.S. designed, performed, and analyzed the experiments. F.M. performed FE-SEM imaging and assisted with CLSM imaging. J.K. was responsible for NMR analysis of lignin with contribution from M.H.S.; A.P. purified and characterized HiC and participated in the design of experiments. J.S. carried out cryo-sectioning and TEM imaging of the alginate beads. M.H.S and M.Ö. co-wrote the manuscript. All authors discussed the results and commented on the manuscript.

## Additional information

**Competing interests:** The authors declare no competing interests.

