## [Peer Review File · Nature Communications]

Reviewers' comments:

Reviewer #1 (Remarks to the Author):

This manuscript reports the fabrication of spatially confined lignin nanospheres for biocatalytic ester synthesis in aqueous media. The composite materials enable aqueous ester synthesis by forming spatially confined biocatalysts upon self assembly and dehydration-driven aggregation in calcium alginate hydrogel. I realized that the authors have paid great effort in studying the catalytic properties of the materials. However, I am not fully convinced by the novelty of this work. Actually, I found the main idea are quite overlapped with the following reports, two of them are indicated by authors in the introduction section. A novel tool for integrated, spatially confined enzymatic reactions. *Angewandte Chemie* 2007, 46 (29), 5605-5608. Enzymatic reactions in confined environments. *Nature Nanotechnology* 2016, 11 (5), 409-420.

Some specific comments:

1. As far as I understand, the lipase was immobilized on the lignin nanospheres by the electrostatic force rather than the covalent bond commonly used in other papers. Is this immobilization stable upon different conditions? Such as pH or ionic solution.
2. The authors confined the lignin nanospheres in the alginate hydrogels that crosslinked by calcium ion. Based on our experience, the calcium crosslinked alginate is not stable enough in PBS solution for a long time (< 5 days), especially in alkaline conditions, the hydrogels would be rapidly break.
3. The authors should test the long time stability of the composite materials (especially for different pH conditons), also the possible expose rate of enzymes should be investigated.
4. How about the reusability of the nanospheres, the authors should provide the cycle performance of their samples.
5. The authors claimed that their products were superior to that of industrial products, however, the reviewer would like to see the comparison with current state-of-the-art research works, since there are numerous lipase immobilization reports.

Reviewer #2 (Remarks to the Author):

This manuscript is a key contribution to the field of catalysis, advancing the use of sustainable bioresources and solving a key challenge in green chemistry (cat. synthesis of esters in water). The manuscript is very well written, the experimental plan and controls fully support the authors' conclusions and it will form a foundation publication for future studies in this field. I full support publication.

Reviewer #3 (Remarks to the Author):

The immobilization of the enzyme in a support in an oriented way and the detailed study of the formation of the bonds, draws attention in this paper. The main novelty of the manuscript is the esterification reaction to be confined, allowing the surrounding medium to be aqueous. Are they novel and will they be of interest to others in the community and the wider field? The quantity and quality of the analyzes can help the area's researchers and should be considered.

Some points should be better explained and discussed. So I suggest major revisions :

- The authors comment on the purification of enzymes, but what is the purification factor? What are the protein concentrations? The authors, in various parts of the text, comment “yield of the biocatalysts...” but what is purity? Can the direct relation be made? How were the enzymes obtained?
- Figures and analyzes should be better introduced and discussed.
- Is it possible to measure the enzymatic activity during the steps of the material's synthesis?
- In the page 16, when the authors mention the amount of immobilized enzyme, how were the calculations performed? Is it a theoretical value?

Dear anonymous referees,

Thank you for your constructive feedback that enabled us to improve the manuscript. It has been carefully revised according to the comments received, as indicated by our detailed responses below.

Reviewers' comments:

Reviewer #1 (Remarks to the Author):

This manuscript reports the fabrication of spatially confined lignin nanospheres for biocatalytic ester synthesis in aqueous media. The composite materials enable aqueous ester synthesis by forming spatially confined biocatalysts upon self assembly and dehydration-driven aggregation in calcium alginate hydrogel. I realized that the authors have paid great effort in studying the catalytic properties of the materials. However, I am not fully convinced by the novelty of this work. Actually, I found the main idea are quite overlapped with the following reports, two of them are indicated by authors in the introduction section. A novel tool for integrated, spatially confined enzymatic reactions. *Angewandte Chemie* 2007, 46 (29), 5605-5608. Enzymatic reactions in confined environments. *Nature Nanotechnology* 2016, 11 (5), 409-420.

We thank the reviewer for the careful reading and constructive suggestions. The paper by Kreft and co-workers describes the synthesis of shell-to-shell calcium carbonate microcapsules filled with magnetite-based nanocapsules, while our manuscript presents the novel synthesis of cationic lignin nanospheres embedded in calcium alginate, therefore having a wholly different structure, chemical composition and properties. The central part of our work was the use of cationic lignin nanospheres as renewable templates to anchor hydrolases, whereas Kreft et al showed matrix-type entrapment of proteins, and did not study hydrolases. In contrast, the main purpose of our work was to tackle one of the most demanding enzymatic reaction in water, i.e. the ester synthesis. While we want to give credit to the earlier publications, and included the paper of Kreft et al in the references, we also wish to point out that our approach of adsorption-entrapment immobilization of hydrolases consists of fewer steps and uses less expensive and biodegradable materials. Overall, our method is conceptually different and holds promise for industrial applicability.

Coming to the paper by Kuechler and co-workers, it is an excellent general review of the topic, but also in this case none of the cited papers present a lignin-alginate system for enzymes encapsulation. Moreover, the material used in our work for producing nanospheres, lignin, an untapped renewable plant polymer, was not considered at all in these previous reports. In the light of these observations, we are confident that our manuscript introduces a significant degree of novelty that should be of interest to the broad readership of *Nat. Comm.*

Some specific comments:

1. As far as I understand, the lipase was immobilized on the lignin nanospheres by the electrostatic force rather than the covalent bond commonly used in other papers. Is this immobilization stable upon different conditions? Such as pH or ionic solution.

Yes, it is correct that we did not use chemical cross-linking, but utilized adsorption and subsequent entrapment. We also investigated Layer-by-Layer assembly as a means of enzyme immobilization, but in that case the ionic strength of the aqueous reaction mixture (150 mM butanol, 50 mM butyric acid) destabilized the structure, making the repeated use of the biocatalyst preparation impossible. However, the developed spatial confinement approach solved this problem as is evident from the successful repeated use of the immobilized enzymes and their excellent stability in water (Fig. 2a,b,d). In response to the reviewer's question, we performed additional experiments checking the stability of the system. New data obtained from the revision experiments provided further support for the stability (Fig. S4, Fig. S6, and Fig. S12b). We believe that the electrostatic interactions were important in orienting the hydrophobic domains of enzymes towards the aqueous solvent environment, hence contributing to the water-repellant stability. The new results further confirm that drying was important, as it rendered the beads durable (Fig. S6) and resistant to protein leaching in high ionic strength solutions (Fig. S12).

2. The authors confined the lignin nanospheres in the alginate hydrogels that crosslinked by calcium ion. Based on our experience, the calcium crosslinked alginate is not stable enough in PBS solution for a long time (< 5 days), especially in alkaline conditions, the hydrogels would be rapidly break.

The initial selection of using calcium alginate as the entrapment matrix was based on its demonstrated excellent stability. The key to success in our work was the drying of the hydrogel that rendered the material resistant to mechanical forces. We conducted up to 5 days of repeated agitation at 200 rpm at 40 °C in the biphasic esterification experiments, yet the catalyst beads remained macroscopically intact. It is true that undried calcium alginate hydrogels tend to break by ion-exchange in PBS buffer (Figure R1). However, in our approach the catalysts were further strengthened by the incorporated cationic lignin nanospheres. After drying this composite structure, the beads were not disintegrated during 72 h magnetic stirring in 0.2 M buffer solutions at pH 2–10.5 (Fig. S6). Most importantly, the beads appeared to be stable at pH 3, appropriate pH for the esterification reaction.

Figure R1 | pH-stability of undried alginate beads. Wet alginate hydrogel beads photographed after 72 h magnetic stirring (120 rpm) in 0.2 M pH buffers.

3. The authors should test the long time stability of the composite materials (especially for different pH conditons), also the possible expose rate of enzymes should be investigated.

We think this is an important point that differentiates our work from prior literature. Our biocatalyst were dry beads and not soft hydrogel beads traditionally stored in wet state or lyophilized to render them extremely fragile. Because pH is not as relevant a factor for storage of dried alginate beads, we opted to determine stability of the beads in aqueous pH buffers from pH 2 to pH 10.5 (Fig. S6). Prior literature has evidenced that there is no leakage of enzymes from calcium alginate beads in organic solvents: ref [15] and references therein. For assessing aqueous stability, we carried out an experiment

that determined leakage of protein from undried hydrogel beads and dried spatially confined biocatalysts (Lipase M with c-CLPs). The results in Fig. S12b show clearly that there is practically no loss of protein from dried beads in contrast to up to 21% loss from the wet hydrogels at alkaline pH 10.5.

The long-term storage stability was assessed by repeating the butyl butyrate synthesis with immobilized HiC (spatially confined with c-CLPs) that had been stored 251 days at 4 °C. The results in Fig. S12a indicate a decay of catalytic activity of 29% during this 8-month period. The decay of catalytic activity was also determined during the ester synthesis in aqueous-organic media (Fig. S8e), and the $t_{1/2}$ values are reported on page 7: “Another benefit offered by our immobilization system is the stabilization of lipases. The time required to lose one half of the catalytic activity ($t_{1/2}$, Fig. S10e) under actual reaction conditions is higher with spatially confined Lipase M (98 h) and HiC (161 h) compared to the literature values of 24–32 h with covalently immobilized lipases.^{27,34} Furthermore, the ability to carry out the reaction in the presence of the 90% volume fraction of water further stabilized Lipase M ($t_{1/2}$ =110 h).”

4. How about the reusability of the nanospheres, the authors should provide the cycle performance of their samples.

The price of the colloidal lignin particles, i.e. the nanospheres, was recently estimated at approximately 1 EUR/dry-kg (Lintinen et al., Green Chem. 2018). At this cost, we reasoned that it is suitable to utilize the catalytic activity until it reaches a certain minimum level. Instead of carrying out a chemically burdensome extraction of the components, we reasoned that the biodegradable catalysts can be disposed of by composting or combustion at the end of their lifetime. However, we remain open for future research that could consider these materials as a possible carbon source for microbial fermentations, adsorbents for water purification, etc.

5. The authors claimed that their products were superior to that of industrial products, however, the reviewer would like to see the comparison with current state-of-the-art research works, since there are numerous lipase immobilization reports.

Thank you for recognizing the significant finding of our work. The unique feature of our approach was to orient the enzymes non-covalently in compartmentalized assemblies that were locked by drying of the hydrogel matrix. This represents a marked conceptual advancement to the current immobilization methods, none of which to our knowledge achieves ester synthesis in aqueous media comparable to our results. We agree that there are numerous papers on lipase immobilization, which is reasonable given their broad industrial and physiological importance. To stay within suitable page limits, and to provide an unambiguous comparison to our results, we selected to focus on the prior works that have used lipases for butyl butyrate synthesis. Overall, the cited references [3–4], [6–11], [15], [17–18], [25–27], [32], [34–35], [37] covers the topic extensively from different angles.

Reviewer #2 (Remarks to the Author):

This manuscript is a key contribution to the field of catalysis, advancing the use of sustainable bioresources and solving a key challenge in green chemistry (cat. synthesis of esters in water). The manuscript is very well written, the experimental plan and controls fully support the authors' conclusions and it will form a foundation publication for future studies in this field. I full support publication.

We thank the reviewer for the positive feedback.

Reviewer #3 (Remarks to the Author):

The immobilization of the enzyme in a support in an oriented way and the detailed study of the formation of the bonds, draws attention in this paper. The main novelty of the manuscript is the esterification reaction to be confined, allowing the surrounding medium to be aqueous. Are they novel and will they be of interest to others in the community and the wider field? The quantity and quality of the analyzes can help the area's researchers and should be considered. **We thank the reviewer for recognizing the important contribution and indicating that this topic deserves attention. We are confident on the novelty and impact of our work, as already discussed above in response to the referee 1.**

Some points should be better explained and discussed. So I suggest major revisions :

- The authors comment on the purification of enzymes, but what is the purification factor? What are the protein concentrations? The authors, in various parts of the text, comment "yield of the biocatalysts..." but what is purity? Can the direct relation be made? How were the enzymes obtained?

Thank you for pointing out these issues. The decision to use purified enzymes was made to generalize the results in a broader context of hydrolases, and to exclude possible effects of impurities and excipients in commercial preparations. To clarify the enzyme purification procedure, the following text was inserted to the Supplementary Methods: "Humicola insolens cutinase (HiC) (Novozymes, China) was received in liquid media and was purified via 3 ultrafiltration steps using a Vivaspin 20 PES twin membrane ultrafiltration unit (Sartorius, Germany) with a cut off of 10K MWCO before further use. The analysis of the purified enzyme via SDS-PAGE showed a 95% purity. These results are in line with a previous report from Feder & Gross that achieved a purity >95% using a Millipore 2000 ultrafiltration cell unit (David Feder, Richard A. Gross, Biomacromolecules 2010, 11, 690–697). After purification the protein concentration, determined via the BSA assay (BioRad protein assay, Bio-Rad Laboratories GmbH, Vienna, Austria) showed a concentration of 12.1 mg/mL."

- Figures and analyzes should be better introduced and discussed.

The manuscript was revised thoroughly to improve its readability. Changes were tracked to enable comparison to the earlier version.

- Is it possible to measure the enzymatic activity during the steps of the material's synthesis?

This is an insightful question. It may prove difficult to provide an unambiguous activity mass balance, because the interfacial activation of lipases increases their activity beyond the "normal" level. Instead, the efficacy of enzyme entrapment can be measured. Hence, we repeated the process and determined protein concentrations before and after the adsorption and entrapment steps. The new results in Fig. S4 show 91% enzyme immobilization during the adsorption step, while the immobilization efficiency reached 96% after the calcium alginate entrapment.

-In the page 16, when the authors mention the amount of immobilized enzyme, how were the calculations performed? Is it a theoretical value?

Thank you for this comment. We believe the new results that were discussed in response to the previous question address this point as well. The values reported in Table S4 are theoretical values. This is now indicated together with the determined protein content (Bradford assay, BSA standards) of Lipase M ($48\pm 3\%$) in the footnote.

In addition to the changes required by the reviewers, additional revisions were limited to minor changes to the ester synthesis percentages. The updated values are marked in the manuscript. These changes were made after careful checking and corrections of the calculations in Excel, and do not affect any conclusions of the work. The raw data used in the calculations has been updated with the new results from the revision experiments and is shared in the Figshare repository.

Reviewers' Comments:

Reviewer #1 (Remarks to the Author):

After carefully reading the revised manuscript, I believed the authors have well addressed all my comments, and the manuscript is suitable for publishing now.

Reviewer #3 (Remarks to the Author):

The authors accept my suggestions, answered my questions and inserted new results and data to clarify my doubts. So I recommend the publication of this article in the Journal Nature Communications.